# Golgi’s Role in the Development of Possible New Therapies in Cancer

**DOI:** 10.3390/cells12111499

**Published:** 2023-05-29

**Authors:** Dragos-Bogdan Vlad, David-Ioan Dumitrascu, Alina-Laura Dumitrascu

**Affiliations:** 1Emergency Clinical Hospital of Saint Pantelimon, 021659 Bucharest, Romania; drvladdragos@gmail.com; 2Faculty of Medicine, Carol Davila University of Medicine and Pharmacy, 050474 Bucharest, Romania; david-ioan.dumitrascu0720@stud.umfcd.ro

**Keywords:** Golgi, cancer dynamics, anticancer therapy, stimulator of interferon genes (STING), glycosylation, protein trafficking, GRASP55

## Abstract

The Golgi apparatus is an important organelle found in most eukaryotic cells. It plays a vital role in the processing and sorting of proteins, lipids and other cellular components for delivery to their appropriate destinations within the cell or for secretion outside of the cell. The Golgi complex also plays a role in the regulation of protein trafficking, secretion and post-translational modifications, which are significant in the development and progression of cancer. Abnormalities in this organelle have been observed in various types of cancer, although research into chemotherapies that target the Golgi apparatus is still in its early stages. There are a few promising approaches that are being investigated: (1) Targeting the stimulator of interferon genes protein: The STING pathway senses cytosolic DNA and activates several signaling events. It is regulated by numerous post-translational modifications and relies heavily on vesicular trafficking. Based on some observations which state that a decreased STING expression is present in some cancer cells, agonists for the STING pathway have been developed and are currently being tested in clinical trials, showing encouraging results. (2) Targeting glycosylation: Altered glycosylation, which refers to changes in the carbohydrate molecules that are attached to proteins and lipids in cells, is a common feature of cancer cells, and there are several methods that thwart this process. For example, some inhibitors of glycosylation enzymes have been shown to reduce tumor growth and metastasis in preclinical models of cancer. (3) Targeting Golgi trafficking: The Golgi apparatus is responsible for the sorting and trafficking of proteins within the cell, and disrupting this process may be a potential therapeutic approach for cancer. The unconventional protein secretion is a process that occurs in response to stress and does not require the involvement of the Golgi organelles. P53 is the most frequently altered gene in cancer, dysregulating the normal cellular response to DNA damage. The mutant p53 drives indirectly the upregulation of the Golgi reassembly-stacking protein 55kDa (GRASP55). Through the inhibition of this protein in preclinical models, the reduction of the tumoral growth and metastatic capacity have been obtained successfully. This review supports the hypothesis that the Golgi apparatus may be the target of cytostatic treatment, considering its role in the molecular mechanisms of the neoplastic cells.

## 1. Golgi’s Major Roles

Camillo Golgi made an important observation while studying Purkinje cells in the cerebellum in 1898 using silver nitrate staining. He identified what would later become known as the Golgi apparatus using this technique and, shortly after, discovered its presence among neural cells of spinal ganglia. Soon thereafter, it was identified in other structures; initially known as an “internal reticular apparatus”, its function remained unclear at that time, and, for an extended period, it was thought to be just a staining artifact [1].

The Golgi apparatus, also known as the Golgi complex or Golgi body, is a cellular organelle present in eukaryotic cells where it fulfills its function in the modification and transportation of proteins and lipids. It is made up of a series of flattened, membrane-bound sacs called cisternae, which are arranged in stacks called Golgi stacks. There are typically between three and eight Golgi stacks in a cell, and each stack can contain five to seven cisternae [2].

One of the primary roles of the Golgi apparatus is to sort and pack proteins for transport to their final destination within the cell or for secretion outside the cell. Proteins that are created in the Endoplasmic reticulum (ER) follow an anterograde pathway through the Golgi apparatus, passing through the cis-, medial- and trans-Golgi compartments (Figure 1).

The Golgi apparatus is also responsible for post-translational modifications (PTMs) of proteins. PTMs are chemical modifications that occur after a protein has been synthesized by the ribosome and are essential for proper protein function, one of the most significant being the glycosylation of proteins. This process involves the addition of oligosaccharides to particular proteins or lipids, resulting in the formation of glycoconjugates (glycoproteins, proteoglycans, glycosphingolipids, etc.). It occurs within the ER and Golgi apparatus, and its primary mechanisms are N-glycosylation and O-glycosylation. N-glycosylation is a process where a carbohydrate chain is covalently attached to the amide group of an asparagine residue (N) in the protein. The initial stages of N-glycosylation commence in the ER, where a pre-formed oligosaccharide is moved from a dolichol lipid carrier to an Asn residue located within the N-X-S/T consensus sequence (where X is any amino acid except proline and S/T is serine or threonine). The oligosaccharide is then modified and further processed as the protein moves through the Golgi apparatus [6,7]. O-glycosylation, on the other hand, is a process in which a carbohydrate chain is attached to the hydroxyl group of a serine or threonine residue (S/T) in the protein. There are various types, including mucin-type O-glycosylation, O-GlcNAcylation and O-fucosylation, that occur in the Golgi apparatus and/or other compartments within the cell [6,8]. Furthermore, there are other types of pathways that belong to the process of glycosylation. Sialylation represents the terminal addition of sialic acid, a negatively charged monosaccharide, to a glycan chain on a protein or lipid. It plays a crucial role in numerous biological processes such as inflammation, immune response and cell–cell recognition. Sialylation mainly occurs in the Golgi apparatus and is facilitated by a group of enzymes known as sialyltransferases [9,10]. In addition, fucosylation is also a biochemical process in which fucosyltransferases (FUT) enzymatically transfer a fucose molecule from GDP-fucose to glycan structures [11]. So, the Golgi apparatus is responsible for the final processing and sorting of glycoproteins, which involves the addition, removal and modification of carbohydrate groups.

In order to proceed with the review, we need to fathom how this organelle interferes when the tumoral cell’s request for proteins increases and overwhelms the capacity of the Golgi in cancer, leading this way to the insufficiency (or stress) of the function [12]. When the organelle is exposed to stress, it triggers a cascade of molecular events that are collectively referred to as the Golgi stress response, a cellular mechanism that is activated when the functionality of the organelle is impaired, aiding in the restoration of its normal function. The pathways of the Golgi stress response, as they are for ER, do not seem very tangled. There is a group of sensor molecules that identify when the Golgi apparatus is not performing its tasks adequately. This, in turn, activates a transcription factor that encourages the transcription of genes related to the Golgi apparatus. The transcriptional enhancer element is where the aforementioned factor binds, and it targets genes that encode Golgi-related proteins, with a noteworthy mention of the glycosylation enzymes [13]. Only five stress response pathways have been described—transcription factor E3 (TFE3), cyclic AMP-responsive element-binding protein 3 (CREB3), proteoglycan (PG), mitogen-activated protein kinase MAPK-ETS and heat shock protein 4 (HSP47)—although many of the other elements remain to be clarified [14]. So, the Golgi stress response is an important cellular mechanism that helps to maintain the function and integrity of the Golgi apparatus in tumoral cells, making it worthy of further studies leading probably to therapies that counter the pathological effects of them. 

Here, we review how dysfunctions of these roles interfere with cancer and highlight examples of therapies to combat tumor progression and survival.

## 2. Golgi Effects in a Cancer Cell Invasion

Firstly, the Golgi apparatus is involved in the secretion of proteins and lipids from the cell; hence, various alterations can affect the composition of the secretome—the collection of all the proteins and other molecules secreted by a cell—and the extracellular matrix (ECM), leading to numerous diseases, including obesity, diabetes and cancer [5]. Several gene-encoding proteins involved in Golgi-associated processes have been found to be frequently mutated in cancer. In addition, mutations in these genes are associated with enhanced metastasis and lower patient survival in several types of cancer, including breast and lung [15]. 

Increased secretion and deposition of ECM components can contribute to cancer development and progression. Matrix metalloproteinases (MMPs) can degrade components of the ECM, allowing cancer cells to invade and migrate through tissues, and can also promote angiogenesis by releasing VEGF and other signaling molecules from the ECM. Furthermore, some MMPs are important for the degradation of the basement membrane, this process being necessary during the metastatic cascade [5,11].

The Golgi apparatus performs a vital function in releasing immune components such as cytokines and chemokines, which take part in the formation of the tumor micro-environment (TME). In many types of cancer, alterations in the Golgi apparatus can lead to an increased secretion of immune factors that promote an immunosuppressive TME, which can contribute to cancer development and progression. For example, Phosphatidylinositol 4-kinase III beta (PI4KIIIβ) is a lipid kinase primarily localized to the Golgi apparatus, where it has been shown to regulate the activity of several signaling pathways that are involved in cancer development and progression [15]. In addition, GOLPH3 (Golgi phosphoprotein 3) and CKAP4 (cytoskeleton-associated protein 4) are two proteins localized in the Golgi apparatus and upregulated in several types of cancer by enhancing the secretion of exosomal WNT3A [16]. Furthermore, inflammatory cells secrete cytokines (TNF-α, IL-6, etc.) which can promote chronic inflammation within the TME, leading to cancer cell proliferation, survival and migration [17].

As mentioned before, the Golgi complex is also responsible for the well-mediated trafficking of the secreted proteins within or outside the cell. Dysregulation of this process in cancer can result in altered protein localization, increased protein secretion and the activation of oncogenic signaling pathways [5,18]. In particular, RAB and ARF proteins (ADP-ribosylation factor) are both involved in the regulation of Golgi-mediated trafficking, but they play distinct roles in this process: RAB proteins are involved in transport from the ER to the Golgi, intra-Golgi and from the Golgi to the other organelles or the plasma membrane, while ARF proteins regulate vesicle formation and budding from membranes. A dysregulation of both GTPases leads to disrupted protein trafficking and contributes to the development and progression of cancer [19,20]. Moreover, SNARE and SNAP proteins, which are critical for the regulation of membrane fusion events in cells, are adjusted in cancer cells to promote invasion [21]. Golgi-specific proteins also can act as oncoproteins when their expression is upregulated or their function is altered in cancer cells—GOLPH3’s increased secretion causes exocytosis of pro-metastatic factors [8,16]. Hence, Golgi’s role in protein trafficking is significant in tumoral cells because of the great number of molecules used for survival of cancer.

Secondly, the orientation of the Golgi can alter its function and contribute to cell polarity and migration in the front–rear polarity model [22]. The Golgi can dynamically reorient itself in response to migratory cues, such as chemokine gradients or extracellular matrix (ECM) stiffness, to guide vesicular flow toward the leading edge of the cell. Out of consideration for this, in migratory cells, this directional flow can facilitate the localized secretion of pro-migratory factors, such as MMPs, growth factors and cytokines, which can promote cell migration and invasion [23]. So, the orientation of the Golgi apparatus regulates exocytosis and directs the movement of vesicles toward the front edge of the cell, promoting directional migration that can facilitate cancer metastasis.

On the other hand, the morphology and dynamics of the Golgi can be altered in cancer, and these changes can contribute to tumor progression and metastasis. In normal cells, the Golgi is typically organized as a compact perinuclear structure with distinct cis, medial and trans compartments. However, in cancer cells, the Golgi can exhibit a range of morphological changes, including fragmentation, expansion and redistribution throughout the cell [11]. It is generally agreed that a well-organized Golgi structure is essential for precise post-translational modification and sorting of proteins. Thus, Golgi fragmentation, usually triggered by cellular stresses, growth factors and mitotic phosphorylation of Golgi structural proteins, accelerates protein trafficking and cell proliferation [24]. 

Aside from the Golgi’s altered morphology and orientation and its influence on the secretome, there is another adaptation that this organelle adopts when it comes to tumoral growths and metastasis. The accumulation of evidence indicates that changes in the glycosylation patterns of carrier proteins—including partial formation of glycan structures, increased expression of complex branched N-glycans and truncated O-glycans (Tn and Sialyl Tn antigen), as well as changes in the expression of sialylated glycans and an increase in the expression of ‘core’ fucosylation—promote the procurement of cellular traits that are essential for the malignant transformation of cells [25]. Changes in glycosylation appear to have a direct impact on cell growth and survival, as well as promoting tumor-associated immune responses and metastasis. 

N-glycosylation is a highly conserved process that occurs via the catalysis of enzyme complexes called Oligosaccharyltransferases (OST), located in the ER membrane. The OST complex operates by binding glycans to specific asparagine sites on newly forming polypeptides within the ER lumen. The mammalian OST complex is composed of various subunits, among which STT3 (STT3A and STT3B) has a crucial function in N-glycosylation processes in mammals. Blocking the STT3A subunit amplifies the unfolded protein response in mammalian cells, leading to detrimental effects such as cancer. Inhibition of OST enzyme activity results in the disruption of proper protein folding in cells, which is a consequence of its high substrate specificity towards glycosylation. This perturbation in the process can lead to various pathological conditions [26]. 

However, the abnormal expression of N-acetylgalactosaminyltransferase 3 (GALNT3), a gene that encodes UDP-GalNAc transferase 3, enhances the O-glycosylation of MUC1. This process leads to increased stabilization of the E-cadherin and β-catenin complex, ultimately stimulating cell migration and proliferation in ovarian cancer cells [27]. Therefore, suppression of cell proliferation and invasion is observed when MUC1 destabilization occurs via GALNT3 inhibition [28]. Moreover, it was shown that the interaction between MUC1 and β-catenin leads to the destabilization of adherent junctions, the disturbance of cytoskeletal architecture and an increased cell invasion in breast cancer cells [8]. MUC16, a transmembrane protein, is involved in promoting cell adhesion contact among epithelial cells via its interaction with the actin cytoskeleton within the extracellular matrix (ECM). Disruption of MUC16 genetic expression interferes with the binding between the actin cytoskeleton and the cytosolic domain of MUC16, thereby leading to increased migration and invasion of epithelial cells [29]. 

Numerous transformed tumor cells exhibit distinctive alterations in their sialylation patterns. It has been reported that pancreatic cancer cells with an increased malignant potential are correlated with the overexpression of sialyl Tn (STn) antigen [30,31,32]. 

Along with this, abnormal fucosylation is associated with multiple cancers. Physiologically, this process is catalyzed by thirteen types of fucosyltransferases (FUTs) of which FUT1–11 are known to catalyze N-linked fucosylation and be located in the Golgi apparatus, while the other two types exist in the ER and catalyze O-linked fucosylation. Among the fucosylated glycans, the most well-known are ABO blood group antigens [33]. There were several reports about pathological manifestations that occurred due to dysfunctions of fucosylation: FUT8 was identified as a prognostic marker in patients with colorectal cancer (CRC); changes in the p53 gene alter the predictive relationships between the expression of FUT8 and the prognosis of individuals with CRC. [34]. In addition, a biomarker for detecting colorectal carcinogenesis was found to be an increase in fucosylation of N-glycans [35]. Furthermore, the association between high levels of FUT6 and colorectal cancer progression has been confirmed [36].

As stated in this review until now, we may assume that the Golgi apparatus has its role in the evolution of many kinds of cancer, so we have to wonder whether this organelle may be the target of cytostatic treatments or not. Henceforth, we will elaborate on a few promising approaches that are being investigated, targeting altered compounds of a tumoral cell’s Golgi body. 

## 3. Golgi Apparatus as a Target for Anti-Cancer Therapy

### 3.1. Targeting Stimulator of Interferon Genes (STING)

In some human cancers, an adaptative immune response indicated by the T cell-inflamed tumor micro-environment phenotype has been observed. Thus began the search for the immune pathways of such endogenous responses. 

It has been shown that a very important role of the Stimulator of Interferon Genes (STING) pathway is as a main event required for IFN production, the activation of the dendritic cells and priming of CD8+ T cells against tumor antigens. STING is a protein that is found in the endoplasmic reticulum (ER) and the ER-mitochondria associated membranes.

The STING pathway is an innate immune response, highly dependent on vesicular trafficking, that senses cytosolic DNA and activates several events through multiple post-translational reactions. The cyclic GMP-APM synthase (cGAS) binds directly to DNA and generates cGAMP. The STING protein is activated by binding with the cGAMP and translocates from ER to ER-Golgi intermediate complex, through the Golgi apparatus, to perinuclear vesicles and then to the nucleus where it induces the expression of target genes such as IFN-beta (Figure 2). The vesicular trafficking is dependent on several factors, and the dysregulation of any of them can lead to the blockage of STING trafficking and IFN-beta gene expression. To prevent chronic cGAS activation, the STING pathway also induces autophagy, helping clear the cytosolic DNA.

Autophagy proteins block the cGAMP production by binding to cGAS, leading to degradation of the cytosolic DNA and negatively regulating STING translocation from the Golgi or its association with TBK 1.

Blocking the components of the pathway that hinder the STING signal transduction could shift the balance in favor of producing type I IFN and assist in promoting an immune response against tumors. Despite not undergoing DNA-damaging therapy, certain tumor cells exhibit the existence of cytosolic DNA, which could potentially function as a trigger for STING pathway activation. Nevertheless, reduced expression of cGAS and STING has been detected in both colorectal carcinomas and human melanomas. While tumors may maintain cGAS function, they may lose their ability to activate STING [37].

During preclinical tests, it was observed that mice with STING deficiency had an impaired cross-priming of tumor antigen-specific CD8+T cells and an accelerated tumor outgrowth. This was due to the absence of IFN-beta gene expression by immune cells that infiltrate the tumor, and the host’s IFN-beta signaling is essential for effective adaptive immune responses against tumors. Further studies are needed to evaluate the implications of these findings for human cancer immunotherapy [38,39].

With these findings in mind, STING pathway agonists have been created and are presently undergoing clinical testing with encouraging outcomes. However, initial data imply that not every patient responds to the treatment, resulting in doubts about the individual mechanisms of response and resistance [37].

In order to develop an effective antineoplastic treatment, the mechanism by which tumor cells are resistant to the STING pathway agonists must be fathomed.

It has been shown in tumoral cells that the Golgi stress response is activated. As specified before, the Golgi stress response augments the organelle’s effects in response to the insufficiency of Golgi function (Golgi stress) using five specific pathways, of which it has been observed that the CREB3 pathway induces pro-apoptosis genes [13]. 

It was stated that the pro-apoptotic proteins and cell apoptosis naturally inhibit the STING pathway; hence CREB 3 pathway of the Golgi stress response could be a target for a novel cancer treatment in addition to other therapies using the STING pathway [40].

### 3.2. Targeting Glycosylation

This procedure has emerged as a potential therapeutic strategy for cancer treatment. As previously mentioned, altered glycosylation patterns are often associated with cancer progression and metastasis, making them an attractive target for chemotherapy. Several approaches are being explored, such as modulating glycan structures and inhibition of glycosylation enzymes [8].

Firstly, altering the glycosylation patterns on the surface of cancer cells can make them more recognizable to the immune system, leading to enhanced immune responses against the cancer cells. This approach consists of the adjustment of glycan structures, and it can inhibit cancer cell growth, survival and metastasis by targeting specific glycan structures involved in these processes.

The cadherin–catenin complex is composed of transmembrane proteins called cadherins, which interact with the extracellular environment, and cytoplasmic proteins called catenins, which interact with the actin cytoskeleton. Together, they form a system found in adherens junctions that helps cells bind. Dysfunction of the cadherin–catenin complex disrupts cellular integrity and cell–cell interactions in cancer cells. Conversely, it has been reported that the elimination of certain N-glycans from E-cadherin increases the interaction of the cadherin–catenin complex, thus promoting the stabilization of cell–cell adhesion. Altering the N-glycan structure at the Asn-554 site by adding β1,6-GlcNAc impedes the biological activity of E-cadherin [41]. Studies indicate that blocking the function of α-1,6 fucosyltransferase enhances the function of E-cadherin in cell binding, thus reducing the invasion of lung cancer cells [8].

Integrins are membrane-spanning proteins that control interactions between cells and the ECM. They play a crucial role in maintaining the cytoskeletal architecture and promoting cell growth and proliferation by associating with ECM proteins. Any changes in the glycosylation pattern of anchoring proteins can lead to the development of various diseases, including cancer. The removal of the N-acetylglucosaminyltransferase III (GnT-III) enzyme from the genes obstructs the creation of a β−1,6 branch of complex N-glycans, which hampers the production of integrins. Consequently, this hindrance leads to the prevention of invasion and metastasis in melanoma cells of mice [8].

The Notch signaling pathway is a family of transmembrane receptors that play a crucial role in cellular communication and development. It consists of Notch ligands and Notch receptors. The latter are predominantly O-glycosylated; hence, there has been reported aberrant expression of Notch-modifying glycosyltransferases in various types of cancers [42]. The silencing of protein O-Glucosyltransferase 1 resulted in decreased Notch activation in human myeloid leukemia U937 cells. Therefore, the glycosylation of receptors and cell surface components plays a significant role in directing oncogenic signaling pathways during the development of tumors. As a result, the oncogenic potential of glycans in cancer could be decreased by focusing on the abnormal glycoforms of glycolipids and glycoproteins [8]. 

Therefore, the glycosylation of receptors and cell surface components plays a significant role in directing oncogenic signaling pathways during the development of tumors. As a result, the oncogenic potential of glycans in cancer could be decreased by focusing on the abnormal glycoforms of glycolipids and glycoproteins. 

Secondly, the usage of small molecule inhibitors targeting glycosylation machinery is being investigated as a potential therapeutic strategy for treating cancer. Most glycosylation inhibitors work by disrupting precursor metabolism or intracellular activities. Inhibitors that target abnormal glycosylation are typically small molecules that can be easily absorbed by different cell types which makes them a potential therapeutic option for the treatment of various diseases. Several inhibitors targeting glycosylation are under investigation in clinical trials as potential therapeutic strategies for preventing the progression of cancer. In particular, in a phase 1/2 clinical trial, the combination of Uproleselan (GMI-1271) and E-Selectin Antagonist with chemotherapy demonstrated high rates of disease clearance and encouraging survival results in individuals with Acute Myeloid Leukemia [43]. 

There are several glycosyltransferases that also play an important role in cancer biology and are suggested as potential therapeutic targets. ALG3 (Alpha-1,3- Mannosyltransferase) was proved to be significantly upregulated in hepatocellular carcinoma, and its inhibition impeded the tumoral proliferation [44], while it also holds promise as a potential marker for increased sensitivity to radiation therapy, serving as an effective target to reduce radioresistance by modulating the glycosylation of TGFBR2 in breast cancer [45]. In addition, studies have shown that suppression of GALNT6 (pp-GalNAc-T6) decreases the oncofetal fibronectin expression but also effectively reverses the multidrug resistance (MDR) phenotype, as evidenced by heightened sensitivity to various types of anticancer medications [46]. Furthermore, the expression of B4GALNT1 (Beta-1,4-N-Acetyl-Galactosaminyltransferase 1) could potentially contribute to the regulation of the tumor micro-environment (TME) and tumor immunity [47]. So, inhibitors of glycosylation machinery have shown potential as therapeutic targets for cancer; still, additional investigation and trials are required to thoroughly assess the effectiveness and safety of glycosylation inhibitors as chemotherapy agents. 

To conclude, by inhibiting glycosylation, the growth, survival and metastasis of cancer cells may be impaired. Such inhibitors can also enhance the immune response against cancer cells by altering the glycan structures on the cell surface and making them more recognizable to immune cells.

### 3.3. Targeting Golgi Trafficking

As already stated, cancer and associated stromal cells secret multiple macromolecules (proteins, lipids, messenger RNA, etc.) in the tumoral micro-environment. The cancer secretome differs considerably from that of normal cells and plays a role in tumoral invasion and metastasis by sustaining cancer stem cells, generating a pro-tumorigenic environment and creating premetastatic niches. Unconventional protein secretion is a stress-induced mechanism that circumvents the Golgi apparatus. Genetic and epigenetic changes in cancer cells, as well as their interactions with the tumor micro-environment, are responsible for the aberrant regulation of secreted factors [48]. 

The p53 gene, which is often mutated in cancer, is a tumor suppressor that disrupts the usual cellular response to DNA damage by interfering with processes such as DNA repair, apoptosis and senescence [49]. The mutant p53-driven upregulation of the Golgi reassembly-stacking protein 55kDa (GRASP55) was indirectly mediated via the inhibition of miR-34a (Figure 3). GRASP55 has a very important role in cell proliferation, colony formation, migration and invasion, as in tumoral growth and metastases [50].

The secretome produces and releases two proteins, insulin-like growth factor binding protein-2 (IGFBP2) and osteopontin (SPP-1), outside the cell through the GRASP55-GOLGIN45-MYOIIA secretory pathway.

Moreover, GRASP55 enhances the stability of the GOLGIN45 (G45) protein by preventing its degradation, and it also boosts the expression of MYOIIA by suppressing miR-34a. GRASPIN is a small molecule inhibitor of GRASPS55, and the GRASPIN treatment reduced the level of SPP-1 and IGFBP2, so it reduced the tumor growth and metastatic capacity in multiple preclinical models [51].

## 4. Conclusions

Cancer is a complex disease that involves numerous cellular changes and can have devastating effects; hence, therapy that combats its effects has become one of the primordial targets of today’s medicine. We gauged the significance of the Golgi complex during the manifestations that occur in cancer: the organelle fulfills a cell’s over request for particles by increasing the synthesis of proteins, altering post-translational modifications and disrupting trafficking of molecules. 

Therefore, the Golgi apparatus deserves some attention when it comes to cytostatic treatment. Furthermore, the presumptions were not wrong; we managed to find some promising approaches that might thwart the ruthless dynamics of cancer and proceed with some helpful remedies. Golgi dynamic regulators, cancer cell proliferation, invasion and metastasis can be inhibited, providing a promising avenue for the development of novel therapies.

## Figures and Tables

**Figure 1 cells-12-01499-f001:**
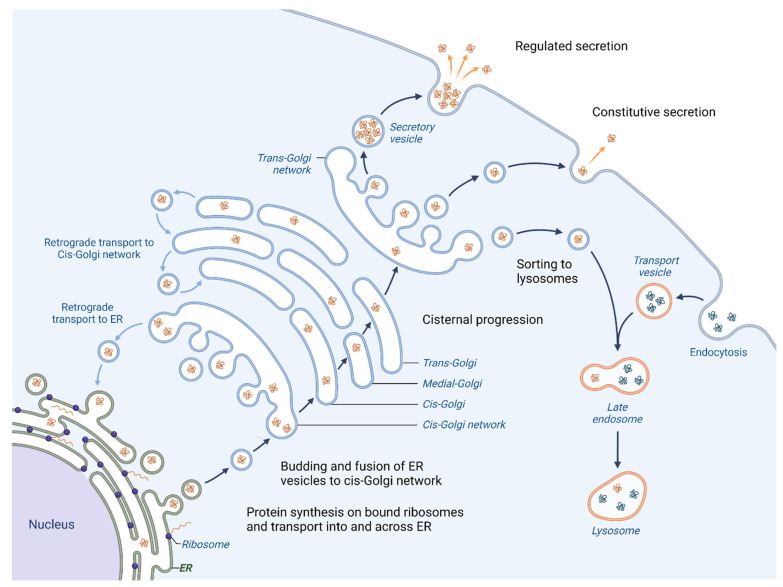
Vesicular trafficking in cell. Initially, the proteins are packaged in vesicles within the ER. A part of them joins together to form the cis-cisterna of the Golgi apparatus [3]. Enzymes within the organelle modify the proteins as they progress through the Golgi, with the specific modifications acting as a signal to determine the final destination of the proteins. The proteins are then sorted within the trans-Golgi network and packaged into vesicles that are transported to specific locations [4]. Certain proteins are directed towards lysosomes for breakdown (proteases, lipases and glycosidases), whereas others are released outside of the cell where they aid in creating the extracellular matrix (ECM), engage with or modify the ECM or facilitate communication with other cells [5]. Moreover, the Golgi apparatus is responsible for retrograde transport of proteins; hence the recycling of proteins back to their origins is possible.

**Figure 2 cells-12-01499-f002:**
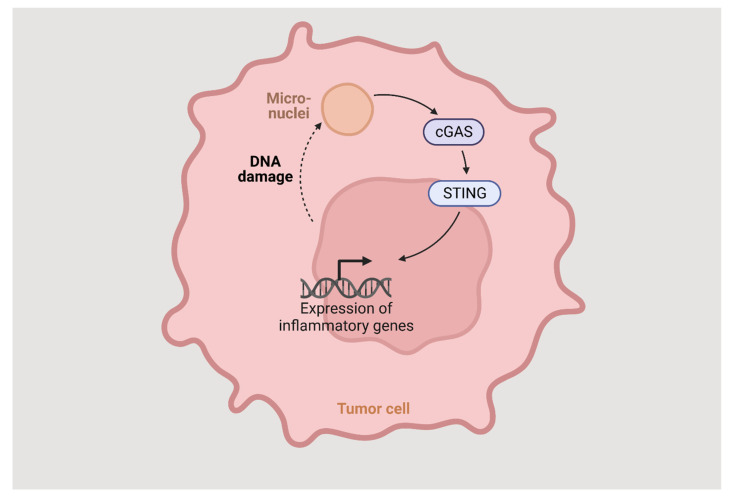
The STING pathway.

**Figure 3 cells-12-01499-f003:**
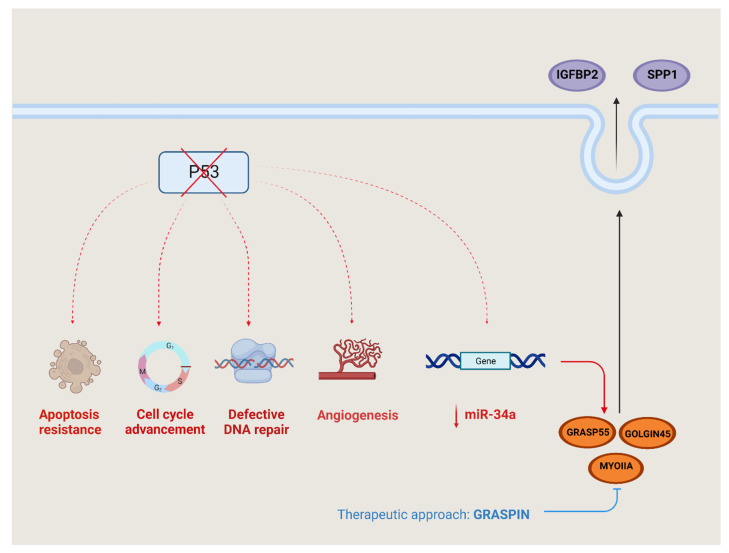
P53 regulation and signaling. Mutated p53 disrupts various pathways within cancer cells, leading to several detrimental effects—resistance to programmed cell death (apoptosis), evading of the normal checks in the cell cycle, impaired DNA repair mechanisms and the stimulation of new blood vessel formation (angiogenesis). Additionally, there is an intricate process involving the extracellular release of certain substances controlled by the suppression of miR-34a expression—insulin-like growth factor binding protein-2 (IGFBP2) and osteopontin/secreted phosphoprotein 1 (osteopontin/SPP-1. The release is facilitated by the proteins such as Golgi reassembly-stacking protein 55 (GRASP55), GOLGIN45 and myosin IIA (MYOIIA). GRASPIN hinders the binding between GRASP55 and GOLGIN45, effectively impeding the functioning of this secretory pathway. As a result, the secretion axis can be inhibited through the use of GRASPIN.

## Data Availability

No new data were created or analyzed in this study.

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
