# Peer review of "Golgi’s Role in the Development of Possible New Therapies in Cancer"

_cells, 2023, doi:10.3390/cells12111499_

Round 1
Reviewer 1 Report
Summary:
In this review article, Vlad et al., describe in detail the Golgi apparatus, which is known to act as an important organelle of eukaryotic cells. The authors supports the hypothesis that the Golgi apparatus might be the target of cytostatic treatment, considering its role in the molecular mechanisms of the neoplastic cells. Overall, the manuscript is well written, and organized. However, I do have some remarks prior to acceptance for publication.
Major Concerns:
- The second paragraph is long, and contains a lot of information, but has only two references. I suggest the authors add some references throughout the paragraph, mainly at the beginning of the paragraph, when the authors begin to describe the compartments of the ER (PMID: 36123032, PMID: 33852913, PMID: 34912111).
- In the third paragraph, the authors addess important types of glycosylation (O- and N-glycosylation), but cite only one paper from 2016, which focuses on the role of glycosylation in health and disease. I suggest the authors also cite a well known paper published by Dr. Spiro (PMID: 12042244), which addresses the different types of glycosylation in mammalian cells.
- In my opinion before discuss about glycosylation, the authors would argue, even briefly, about the metabolic reprogramming adopted by the transformed cells (PMID: 27634447). As the authors may know, the greater uptake of glucose by cancer cells, combined with the differential expression of glycosyltransferases, contribute to the transformed cells stand an aberrant glycophenotype when compared to cells from healthy tissues. This type of information is important to guide the reader on this complex and important subject in oncobiology. Today, the unusual glycosylation in transformed cells is already considered a hallmark of cancer (PMID: 27007155).
- The authors mention throughout the text some glycoproteins, which are used as biomarkers and also as potential therapeutic targets in the treatment of cancer. In addition, they discuss some glycosyltransferases that are also identified as molecular targets in the fight against cancer. This subject is quite interesting and discussed in the literature. Over the past 10 to 15 years, many research groups have focused their studies on this promising area in oncobiology. The authors talk about GNT-V, which is responsible for the biosynthesis of tri/tetra-antennary N-linked glycans. They also discuss GALNT3 (pp-GalNAc-T3) and sialyltransferases. However, authors should discuss the role of other glycosyltransferases and/or their products that also play an important role in cancer biology, and are suggested as potential therapeutic targets. Among these, I suggest the authors describe about ALG3 (ALG3 Alpha-1,3- Mannosyltransferase) (PMID: 35782861, PMID: 33931075 ), GALNT6 (pp-GalNAc-T6) (PMID: 27659430, PMID: 36882122) and B4GALNT1, Beta-1,4-N-Acetyl-Galactosaminyltransferase 1 (PMID: 3198829, PMID: 35935585). These glycosyltransferases and/or their products (unusual glycoconjugates) in transformed cells, in addition to promoting cancer progression via the activation of the metastatic cascade, are also involved in the acquisition / maintenance of the MDR phenotype, one of the major obstacles faced in cancer treatment. The authors must mention these enzymes, which are known to modulate the behavior of the cancer cells, and have been extensively studied in recent years. It will certainly enrich and improve the quality of the manuscript. I suggest adding these therapeutic targets in the topic "Targeting glycosylation". It makes more sense when you read the paper.
- The authors address the participation of Golgi in cancer cell invasion. I would like to know if there is already any paper in the literature that correlates the participation of the Golgi in tumor metastasis and/or in the activation of molecular pathways associated with the epithelial-mesenchymal transition (EMT) process. Please discuss this issue in the topic "GOLGI EFFECTS IN CANCER CELL INVASION".
- I missed figures / tables in this review article. I think the authors should add a figure and / or table to improve the quality of the work.
Minor Concerns:
- Please add topic numbering throughout the manuscript. This facilitates reading and also the review process.
- The level of English language is generally good: the manuscript is comprehensible and truly interesting. Nevertheless, proofreading by a native English speaker would be recommended.
The level of English language is generally good: the manuscript is comprehensible and truly interesting. Nevertheless, proofreading by a native English speaker would be recommended.
Author Response
[Reviewer 1]:
- Comment: The second paragraph contains only two references.
Response: Thank you for your feedback on the second paragraph. We appreciate your observation regarding the length and information provided. In order to ensure the accuracy and credibility of the information presented, we agree that adding references throughout the paragraph would be beneficial.
- Comment: The third paragraph also contains few references.
Response: Thank you for your valuable feedback on the third paragraph. We appreciate your input regarding the types of glycosylation discussed and the need for additional references. We agree that citing a well-known paper by Dr. Spiro (PMID: 12042244) would be a valuable addition to the paragraph.
- Comment: We were advised to include in our manuscript information about the metabolic reprogramming adopted by transformed cells.
Response: Thank you for your insightful commentary regarding the discussion on glycosylation. We appreciate your suggestion to include information about metabolic reprogramming adopted by transformed cells before delving into the topic of glycosylation. We agree that providing a brief overview of this aspect would enhance the reader's understanding of the subject.
- Comment: The reviewer suggests mentioning additional enzymes suggested as potential therapeutic targets.
Response: Thank you for your insightful commentary on the topic of glycosylation in cancer biology. We appreciate your suggestions for expanding the discussion to include other glycosyltransferases and their products that play important roles in cancer biology and are suggested as potential therapeutic targets. We agree that addressing these additional enzymes would enhance the comprehensiveness and relevance of the manuscript. In our revised version, we have incorporated the information you provided about ALG3 (ALG3 Alpha-1,3-Mannosyltransferase), GALNT6 (pp-GalNAc-T6), and B4GALNT1 (Beta-1,4-N-Acetyl-Galactosaminyltransferase 1).
- Comment: The reviewer requests information about any existing literature that correlates Golgi involvement with tumor metastasis and the activation of molecular pathways associated with the epithelial-mesenchymal transition (EMT)
Response: Thank you for your comment regarding the participation of the Golgi apparatus in cancer cell invasion! In response to your inquiry, we have conducted a thorough literature search to identify relevant studies on this topic. We have found several papers and incorporated references to key articles that discuss the role of the Golgi apparatus in tumor metastasis and its influence on molecular pathways related to EMT.
- Comment: We were advised to add figures and/or tables to our manuscript.
Response: We appreciate your suggestion to include figures and/or tables to enhance the quality of the work. Visual aids can indeed be valuable in presenting information and improving the overall understanding of the topic. In response to your suggestion, we have carefully considered the inclusion of figures and/or tables to supplement the text.
Reviewer 2 Report
This is an interesting review summarizing findings that provide a potential role of Golgi in anti-cancer therapies. The article would be of much better appeal and interest to the Reader if it was accompanied by figures indicating Golgi's function and its role in biological processes like STING activation, glycosylation and trafficking.
Author Response
[Reviewer 2]
Comment: The reviewer also advices us to add figures and/or tables to our manuscript.
Response: We value your suggestion to augment the quality of our work by incorporating figures and/or tables. Visual aids are recognized for their ability to effectively present information and enhance comprehension of the topic. In light of your suggestion, we have thoroughly assessed the potential inclusion of relevant figures and/or tables to complement the text.
Reviewer 3 Report
This review provides a comprehensive discussion of the general function of the Golgi apparatus, including secretion, intracellular trafficking, Golgi targeting, Golgi structure, glycosylation, and sugar-modifying enzymes in cancer. STING, glycosyltransferases and GRASPs are also discussed as potential anti-cancer targets. Considering that the review is intended for a special issue on the Golgi, it may be beneficial to minimize general descriptions of the Golgi and focus more on its correlation with cancer treatment.
The identified anti-cancer targets have long been recognized as cancer markers. A more specific explanation of how these markers can function as anti-cancer targets would strengthen the discussion.
My specific comments are as follows:
1) The article states that “there are typically between three and eight Golgi stacks in a cell.” However, the reference [1] states that a single stack contains 5 to 7 cisternae. This needs to be clarified.
2) The description that "certain proteins are targeted to lysosomes for degradation" is accurate but lacks specificity. Proteins that are targeted to and function in lysosomes should be mentioned in this context.
3) Reference [3] appears to be a methods paper or protocol rather than provide conceptual information. It would be helpful to cite more appropriate papers or reviews here.
4) What does the term "eight pathways" refer to when discussing processes within the ER and Golgi apparatus?
5) The statement that O-glycosylation is not well-defined and has different types, including mucin-type O-glycosylation, O-GlcNAcylation, and O-fucosylation, needs clarification. Are these not definitions of types of O-glycosylation?
6) The phrase " the process generally known as the end of the glycosylation is referred to as fucosylation" is vague. Does it refer to terminal fucosylation? Remember that most fucosyltransferases are localized to the early Golgi, whereas most sialyltransferases are localized to the late Golgi.
7) The phrase "In order to proceed with the report" should be revised as this is a review article, not a report.
8) The use of "this complex organelle, the Golgi, this organelle, the complex..." detracts from readability. Please rephrase for clarity.
9) The statement "when the tumoral cell’s request for proteins increases and overwhelms the capacity of the Golgi" suggests that Golgi stress occurs after ER stress. Please confirm this.
10) The paragraph discussing the secretome of cancer and associated stromal cells seems repetitive as it partially repeats an earlier section of the manuscript. Please consider revising for originality and flow.
11) Finally, for clarity, "G45" should be referred to as "GOLGIN45" in the statement "GRASP55 increases the stability of the G45 protein by preventing its degradation".
C) The first section describing general Golgi functions could benefit from a comprehensive revision.
8) The use of "this complex organelle, the Golgi, this organelle, the complex..." detracts from readability.
Author Response
[Reviewer 3]
- Comment: In our manuscript, we stated that 'there are typically between three and eight Golgi stacks in a cell,' despite the reference cited afterward contradicting this claim.
Response: Thank you for your suggestion, it was addressed.
- Comment: In the description below the first figure it should be mentioned which proteins are targeted to lysosomes for degradation.
Response: We considered the inclusion of the specific proteins.
- Comment: Reference [3] isn’t exactly about the topic of the previous sentence.
Response: We cited a more appropriate paper here.
- Comment: The sentence about the eight pathways of O-Glycosylation isn’t comprehensible.
Response: We managed to make it clear.
- Comment: The sentence about the different types of O-Glycosylation isn’t that clear.
Response: We modified it so it could be easily understandable.
- Comment: The phrase describing the process of fucosylation isn’t right (‘The process generally known as the end of the glycosylation is referred to as fucosylation’).
Response: We adjusted it.
- Comment: The phrase ‘In order to proceed with the report’ isn’t right since this is a review article, not a report.
Response: Thank you for your suggestion, we changed that.
- Comment: The use of ‘this complex organelle’; ‘the Golgi’ ; ‘this organelle’ should be rephrased for clarity.
Response: We used other words (‘the Golgi apparatus’ / ‘the Golgi complex’ / ‘the Golgi organelle’) for clarity
- Comment: The statement ‘When the tumoral cell’s request for proteins inceases and overwhelms the capacity of the Golgi’ suggests that Golgi stress occurs after ER stress.
Response: We made it clear.
- Comment: The paragraph discussing the secretome of cancer is almost similar to another section in the earlier section of the manuscript.
Response: We considered the suggestion and altered the structure, so it won’t seem repetitive.
- Comment: In the manuscript the protein called ‘GOLGIN4’5 is only reffered to as its abbreviation, ‘G45’ (‘GRASP55 increases the stability of the G45 protein by preventing its degradation’).
Response: We changed it into ‘GOLGIN45’, so the phrase is more clear.
Round 2
Reviewer 1 Report
I thank the authors for clarifying my questions. In this new version, all my comments have been properly addressed, and in my opinion the manuscript is suitable for publication.
Author Response
Thank you very much! We truly appreciate your dedication and commitment to providing a comprehensive review. Your suggestions and recommendations have undoubtedly strengthened the manuscript, and we are grateful for the opportunity to benefit from your expertise.
Reviewer 3 Report
I do not think the following description is correct as the first comment. Neither Ref 1 nor Ref 2 describe such things. If the authors think this is correct, please cite the appropriate reference, or correct the following description as my comment from the first time around. "There are typically between three and eight Golgi stacks in a cell, and each stack can contain any where from a few to several dozen cisternae [1, 2]."
Do not ignore my comment from the first time around that "The identified anti-cancer targets have long been recognized as cancer markers. A more specific explanation of how these markers can function as anti-cancer targets would strengthen the discussion. "
No comments
Author Response
Thank you very much! We truly appreciate your dedication and commitment to providing a comprehensive review. Your suggestions and recommendations have undoubtedly strengthened the manuscript, and we are grateful for the opportunity to benefit from your expertise.
Firstly, we adjusted the paragraph about Golgi's components. We mentioned that each stack of Golgi contains five to seven cisternae, as Li J. et al. stated in their report.
Secondly, we added more explanations about the functionality of some cancer markers, such as the ones who influence the process of fucosylation. We recognize that the connection between these markers and their potential as anti-cancer targets may not have been adequately elucidated in the original version of the manuscript.
Once again, we sincerely thank you for your valuable commentary, which has provided us with an opportunity to improve the clarity and depth of our work.
Best regards,
The authors.
Round 3
Reviewer 3 Report
No comments.
No comments.